# Lung Cancer: Spectral and Numerical Differentiation among Benign and Malignant Pleural Effusions Based on the Surface-Enhanced Raman Spectroscopy

**DOI:** 10.3390/biomedicines10050993

**Published:** 2022-04-25

**Authors:** Aneta Aniela Kowalska, Marta Czaplicka, Ariadna B. Nowicka, Izabela Chmielewska, Karolina Kędra, Tomasz Szymborski, Agnieszka Kamińska

**Affiliations:** 1Institute of Physical Chemistry, Polish Academy of Sciences, Kasprzaka 44/52, 01-224 Warsaw, Poland; mczaplicka@ichf.edu.pl (M.C.); anowicka@ichf.edu.pl (A.B.N.); kkedrakrolik@ichf.edu.pl (K.K.); tszymborski@ichf.edu.pl (T.S.); 2Department of Pneumonology, Oncology and Allergology, Medical University of Lublin, Jaczewskiego 8, 20-090 Lublin, Poland; izabela.chmielewska@umlub.pl

**Keywords:** pleural effusion, surface-enhanced Raman spectroscopy, partial least square, cancer

## Abstract

We present here that the surface-enhanced Raman spectroscopy (SERS) technique in conjunction with the partial least squares analysis is as a potential tool for the differentiation of pleural effusion in the course of the cancerous disease and a tool for faster diagnosis of lung cancer. Pleural effusion occurs mainly in cancer patients due to the spread of the tumor, usually caused by lung cancer. Furthermore, it can also be initiated by non-neoplastic diseases, such as chronic inflammatory infection (the most common reason for histopathological examination of the exudate). The correlation between pleural effusion induced by tumor and non-cancerous diseases were found using surface-enhanced Raman spectroscopy combined with principal component regression (PCR) and partial least squares (PLS) multivariate analysis method. The PCR predicts 96% variance for the division of neoplastic and non-neoplastic samples in 13 principal components while PLS 95% in only 10 factors. Similarly, when analyzing the SERS data to differentiate the type of tumor (squamous cell vs. adenocarcinoma), PLS gives more satisfactory results. This is evidenced by the calculated values of the root mean square errors of calibration and prediction but also the coefficients of calibration determination and prediction (R2C = 0.9570 and R2C = 0.7968), which are more robust and rugged compared to those calculated for PCR. In addition, the relationship between cancerous and non-cancerous samples in the dependence on the gender of the studied patients is presented.

## 1. Introduction

Cancer is one of the most common causes of death (following cardiovascular diseases) and one of the most critical barriers to increasing life expectancy in the 21st century [1]. The World Health Organization (WHO) estimated that in 2019 over 50% of countries (112 of 183 counties in the world), cancer was the first or second cause of death before the age of 70 years [2]. Although the population of Europe represents 9% of the world population, in 2012, one-quarter of cancer cases occurred in Europe [3]. International Agency for Research on Cancer (IARC) GLOBACAN valuated that in 2020, there will be 19.3 million new cases of cancer and 10 million deaths worldwide [4]. In Europe, there was an estimated 4.4 million new cases of cancer and 1.9 million deaths [4]. Among them, lung cancer was the second most frequently diagnosed and first cause of death worldwide [1]. The average five-year survival rate of lung cancers is 18.6 percent lower than many other leading cancer types. The high mortality is due to late diagnosis of lung cancer; therefore, finding a diagnostic method of tumor detection is of great importance.

A pleural effusion is an accumulation of extra fluid in the space between the lungs and the chest wall (pleura). Pleural effusion (PE) can be caused by infections (pneumonia, tuberculosis), liver or kidney disease (cirrhosis), congestive heart failure, autoimmune disorders (lupus, rheumatoid arthritis), and/or pulmonary embolism or trauma [5,6]. Malignant pleural effusions (MPEs) of carcinomas, especially of the lung, breast, and lymphomas, indicate the advanced stage of the cancer disease or disease progression [7]. MPEs could also be a signal for almost all cancer types [8] of spread or that the cancer has metastasized to other areas of the body [9]. Correct differentiation between benign (BPEs; non-malignant) and malignant pleural effusions is still a significant challenge and of great interest due to its influence on proper patient treatments. Patients may benefit from individualized management targeted at treating the underlying disease process and direct control of the fluid. Generally, nowadays, analysis of the pleural effusion is mainly based on such techniques as biochemistry, microbiology, cytology, and heparin-coated syringe for the pH measurement. However, all those techniques carry a small accuracy; e.g., single tumor markers have low sensitivity (<30%). In comparison, the accuracy of cytological examinations of malignant pleural effusion is around 60% (range of 40–87%) and depends on broad factors, such as the presence of cells, tumor type, and the skills of cytologists [10]. The pleural fluid cytology is the most straightforward and definitive method to diagnose MPE. Still, diagnosis cannot be made solely based on the cytology; thus, to obtain a high sensitivity of 90% diagnostic yield for pleural fluid cytology, three methods, namely computer tomography, PET CT, and cytology, need to be combined [11]. Similarly, to confirm MPE etiology in the case of a negative cytological examination, other non-invasive imaging methods are required, e.g., X-ray, ultrasonography, computer tomography, magnetic resonance imaging, and positron emission tomography [7]. The final diagnosis is not straightforward, and therefore, it is very important to establish new, faster techniques with more relayed results. This is especially important for currently used immunotherapy treatments. Therefore, the new approach for diagnosis PE with the spectroscopic method is of great interest.

In this light, the spectral differences between normal, lung cancer, and tuberculous cells [12] (connected to the reduction of glycogen and increased hydrogen bonding of phosphodiester groups of nucleic acid in lung cancer) but also malignant pleural mesothelioma from lung cancer and benign pleural effusion from pleural fluids [13] using Fourier transform-infrared spectroscopy were already shown. One of the promising methods in diagnosis is surface-enhanced Raman spectroscopy (SERS), which already is a powerful technique to study nucleic acids and proteins [14], therapeutic agents [15], drugs and trace materials [16], microorganisms [17], and cells [18]. The most notable recent advances in Raman and SERS include innovative applications as bimolecular sensors for clinical diagnosis of various diseases, such as Alzheimer’s or Parkinson’s [19], and various cancer diseases, such as gastrointestinal [20,21,22,23], skin [24,25,26,27], breast [28,29,30,31], lung [32,33,34,35] (also from saliva [36,37]), and also brain [38,39,40,41,42] cancer.

In SERS spectroscopy, such problems as band overlaps, linearity, and interactions exist. To overcome it, the factors or analysis-based components and multivariate calibration methods have already been successfully and extensively used in the quantitative analysis of spectroscopic data [43,44,45,46,47]. PCR studies only input variables X (spectral information) to summarize the dataset, while PLS considers the relationship between all inputs; thus, X and output variables Y (spectral and concentration data) are analyzed [48,49,50]. The SERS-based method combined with multivariate calculation was already used to detect mutation of epidermal growth factor receptor of malignant pleural mesothelioma from patients with lung adenocarcinoma (90.7% accuracy) [51]. The difference between benign and malignant of the pleural effusion associated with lung adenocarcinoma were investigated using TiO_2_ photo-catalyzed Ag NPs (sensitivity and specificity over 90%) [52]. X. Chen et al. showed that SERS serum c in combination with generic algorithm-LDA and support vector machine (SVM) can achieve diagnosis and staging of diffuse large B-cell lymphoma (accuracy for LDA = 71.1% and SNV = 78.4%) [53] and discrimination of multiple myeloma (accuracy for LDA = 75.5% and SNV = 86.6%) [54]. Moreover, our previous work proved that it is possible to determine the difference between healthy and tumor salivary glands’ homogenates (accuracy 98%) [55].

Herein, the surface-enhanced Raman scattering technique combined with principal component analysis as an optical spectroscopy method, with higher sensitivity and chemical specificity than that in conventional Raman spectroscopy, have been used to maximize the variations between the analyzed groups of cancerous and non-cancerous pleural effusion samples. Through this work, the chemometric supervised techniques of classification (partial least squares—PLS) and quantification (principal component regression—PCR) of obtained SERS results were performed. The reported results indicate that fast, multivariate evaluation of the multiple probes is feasible and may allow for broad application of inaccurate cancer diagnosis, risk classification, and therapeutic strategies.

## 2. Materials and Methods

### 2.1. Materials

The human pleural effusion samples were derived from fourteen patients with cancer (tumor pleural effusion). Histopathological type of cancer was confirmed either by histopathological or cytological examination during the routine diagnostic process. Six samples with non-cancerous lung effusion, considered the control, were collected according to standard best practice and ethics and bioethics guidelines. The information of the sample classification based on histopathological and immunohistochemical staining of pleural effusion is included in Appendix A. Informed consent was obtained from all subjects. The obtained pleural fluid was disposed to cytology examination. The other part of the pleural effusion was directly, after collection, placed in the small aliquots (1.5 mL) and deep-frozen at −80 °C.

### 2.2. Surface-Enhanced Raman Spectroscopy (SERS)

SERS measurements were performed by Renishaw inVia Raman System, which is equipped with a 300 mW diode laser emitting a 785 nm line. Briefly, the laser-excitation source passes through the line filter and focuses on the test sample mounted on an XYZ translation stage with a 50× Leica N plane EPI objective lens (numerical aperture 0.75). The focal volume of laser beam in this configuration is 0.5 mm. The scattered Raman signal was recorded by the same lens projecting the beam through a holographic notch filter to block Rayleigh scattering. The grating mesh has 1200 grooves per 1 mm to obtain a visual resolution of 5 cm^−1^. SERS signals were detected by a RenCam CCD 1024 × 256 pixels detector. For each sample, 15 SERS spectra were collected with 8 mW of laser power measured on the sample in mapping mode (3 mm × 3 mm). Then, the spectra were collected with 20 s irradiation and 5 accumulations (total time for one spectrum is 60 s). Each map was collected for 30 min. Platforms for SERS analysis were prepared according to the already published procedure [56]. The silicon wafer p-type doped with crystal orientation <100> was mechanically cut into a 3 mm × 3 mm fragments and subjected to laser ablation with femtosecond laser (λ = 1030 nm). The laser working parameters were: pulse width 300 femtosecond, repetition rate 300 kHz, distance between scanning lines 30 μm, and scanning rate of the laser beam on the surface of silicon 1.5 m/s. The surface was modified with two ablation layers perpendicular to each other. The thawed sample was directly applied to the active SERS substrate (5 µL),and left to dry at room temperature in the laminar chamber.

### 2.3. Data Analysis

Before multivariate analysis, the spectra were prepared using concave rubber-band baseline correction (number of iterations: 10, number of baseline points: 64 using smoothing) with Savitzky Golay Filter: 9 points OPUS 7.2. software (Bruker Optic GmbH, Leipzig, Germany). Additionally, normalization was performed using the standard normal variable (SNV), and then, Principal Component (PCR) and Partial Least Squares (PLS) as well as regression and Partial Least Squares Discriminant Analysis (PLS-DA) were performed using Unscrambler software (CAMO AS software, version 10.4, Oslo, Norway). PLS was introduced to the regression tasks, and then, the PLS-DA discriminant analysis was performed. The PLS-DA algorithm was used for predictive and descriptive modeling as well as for discriminant selection of variables. Thus, throughout the manuscript, the PLS as well as PLS-DA data are presented accordingly. For PCR the NIPALS and for PLS the Kernel algorithms, both with random cross-validation method of 20 segments, were used.

Additionally, the root mean square errors of calibration and prediction (RMSEC, RMSEP), the coefficients of determination of calibration and prediction (R2C, R2P), and the receiver operating characteristic (ROC) as well as the correlation accuracy (AUC the area under the ROC curve) were determined.

## 3. Results

Generally, fluid builds up in the pleural space due to an overproduction of fluid or/and decreased fluid absorption. If the effusion is due to cancer cells in the fluid, it is called a malignant pleural effusion (MPE). Malignant pleural effusions are present in 7–15% of patients with lung cancer, 25% of malignant effusions being due to breast cancer and 10% accounting for lymphomas, including Hodgkin’s disease and non-Hodgkin’s lymphoma. It should be mentioned that any epidemiological data should be interpreted with caution, as they may show considerable local variability due to, e.g., ethnicity, the local burden of various diseases, the age structure of the population, and the availability and quality of the healthcare system [57]. Adenocarcinoma is the most common histological type for malignant pleural effusion with an unknown primary tumor. Other causes of non-malignant effusion, called benign (BPE), are very common in several non-malignant pathologies, such as decompensated heart failure and following coronary artery bypass grafting.

Firstly, to find the origin of pleural effusion, the samples were cytologically studied. Based on the characteristic microscopic images, the samples were classified as benign and malignant. The diagnostic yield of pleural fluid cytology depends on several factors, such as the extent of disease, the primary site of malignancy, and the histological type of malignancy (adenocarcinoma is commonly diagnosed compared to squamous cell carcinoma). The high rate of diagnostic yield in adenocarcinomas is due to the fact that it desquamates easily in the pleural cavity [58]. On the other hand, squamous cell lung cancer is often more central, where histological samples of the cancer tissue are the primary source of diagnosis. The images presenting the pleural fluid of adenocarcinoma and the histopathological staining of lung cancer tissue of squamous cell carcinoma are gathered in Figure 1.

Differentiation between benign and malignant pleural effusion allows for early treatment of BPE, which decreases the possibility of disease complications. In contrast, early treatment of MPE may increase the quality of life and survival of patients with advanced malignant diseases. Below, the model of differentiation between benign and malignant pleural effusion fluid based on the SERS data and both PCR and PLS analysis, elaborated upon for clinical samples, are presented.

Studied sample classification obtained from histopathological and immunohistochemical staining is gathered in Appendix A. Samples were divided into analyzed sets:
-Non-carcinoma and carcinoma samples causing pleural effusion;-Type of neoplastic illness (squamous cell and adenocarcinoma samples).

The resultant averages of SERS spectra gathered for cancerous and non-cancerous pleural effusion samples are presented in Figure 2. The standard deviation (SD) for bands up to a wavelength of about 1500 cm^−1^ is from 8% to maximum 15% and above 1500 cm^−1^ is up to 25%. Additionally, Appendix A presents the SERS spectra averages from the SERS map of adenocarcinoma pleural effusion calculated for 15 spectra. The standard deviation on the average plots is visualized by the grey color and for the band at 1140 cm^−1^ is 15%. As can be seen within one sample, the spectra are very similar to each other. Therefore, the multivariate method is necessary for further analysis, as is done within this work.

In Figure 2, the main differences among the spectra are mostly due to the bands’ intensities. However, in the spectrum of cancerous samples, new bands were revealed at 853 cm^−1^, 1079 cm^−1^, 1174 cm^−1^, 1223 cm^−1^, 1339 cm^−1^, 1555 cm^−1^, and 1586 cm^−1^. Those bands are assigned to the vibration of proteins, amide III, and guanine in DNA/adenine/TRP (protein). The increased-intensity 1079 cm^−1^ band was already observed in samples derived from cancer patients [38,59]. These changes are most probably due to higher nucleic acid bases in serum caused by the abnormal metabolism of DNA and RNA in cancer patients. Unexpectedly, the squamous cell cancer SERS spectra, the most intensive band, appear at 1397 cm^−1^. This band might be connected with a biologically active product such as carbon monoxide (CO). The CO is produced during the oxidation of heme catalyzed by the heme oxygenase-1 (HO-1). Recently, a growing body of evidence indicates that the HO-1 activation may play a role in carcinogenesis and can potentially influence the growth and metastasis of tumors [60]. The tentative assignment of all bands observed in the SERS spectra is presented in Table 1.

However, all observed changes in the SERS spectral patterns are not sufficient for discrimination between the cancerous and non-cancerous origin of pleural effusion, and as such, this excludes the possibility to use those SERS spectra directly as a medical indicator for tumor detection. Therefore, further analyses were performed over the gathered SERS data using the multivariate method (MVA). Firstly, the principal component regression (PCR) was applied over the SERS data, and then, to more strength categorization and quantification of analyzed data, the partial least square (PLS) method was used. Both methods, PLS and PCR, are used to find the best discrimination algorithm to classify the obtained SERS spectra as coming from non-cancerous or cancerous samples as well as from different types of cancer (squamous cell or adenocarcinoma). The main difference between both methods is the validation set considered in the calculation, which influences the possibility of considering the random noise in the PLS method. Therefore, validation is necessary for the obtained results. Usually performed validation, similar to that used herein, divides the data table into a calibration set and a prediction set. To evaluate the accepted models, the root mean square errors of calibration and prediction (RMSEC, RMSEP) and the coefficients of determination of calibration and prediction (R2C, R2P) are determined. Additionally, the sensitivity and the specificity based on the receiver operating characteristic (ROC) and the correlation accuracy (AUC the area under the ROC curve) are calculated and presented.

Herein, all SERS data were divided into a calibration set and a prediction set in approximately 2:1 ratio (a calibration set—14 samples; and a validation set—6 samples; around 15 spectra for each sample, total 300 spectra). The number of components or factors are selected by the algorithm implemented in Unscrambler as the most important for presented differentiation. The calculated PCR and PLS scores in 2D and 3D scatter plots are shown in Figure 3. The rest of the PCs or Fs important for calculated discrimination are presented in Appendix A, respectively. It should be mentioned that the average of the standard deviation (SD) variations up to 15% (Appendix A) were obtained for the recorded spectra. Therefore, all gathered spectra, recorded for one sample in the mapping mode, were analyzed and included in the presented classification.

Let us consider results of PCR and PLS calculated for non-cancerous and cancerous samples. In the presented PCR data, PC-1 explains 48% of the variance in block X, while PC-2 explains 18%, and PC-3 explains only 8% (Figure 3A). In comparison, PLS data for Factor-1 explain 28% of the variance in block Y with 33% of the spectral data (X matrix), while F-2 explains 26% of the variance in block Y with 24% of analyzed data. The last F-3 explains only 14%, with 13% of the data within X matrix (Figure 3C). Thirteen consecutive components calculated for PCR explain 96%, while only ten factors as the most important for PLS explain 95 % of the variance between studied groups. However, the PCR and PLS data after RMSEC and RMSEP together with R2C and R2P evaluation most significantly prove the advantage of PLS over PCR method. For PCR, RMSEC = 0.1958 and RMSEP = 0.2058, R2C = 0.7967, and R2P = 0.7770; for PLS model, RMSEC = 0.0901 and RMSEP = 0.1289, R2C = 0.9570, and R2P = 0.9129 (see Appendix A). The smaller the value of RMSE, the higher the quality of the quantitative model and the opposite: the smaller the value of RMSE, the stronger the predictive ability of the model achieved. On the other hand, if the predictive values of the coefficients of calibration determination and prediction in square roots are close to 1, they will ensure the correct classification. Therefore, considering the obtained values, the PLS model is much more robust with higher predictive ability.

Next, to find the determination model for SERS spectra gathered for two different types of cancer, namely squamous cell and adenocarcinoma, again, the PCR and PLS were considered (Figure 3B,D). Evidently, for the first view, the PLS gives scores that allow unambiguous assignment of a given cancer type based on the SERS spectrum. Moreover, the RMSE and R2 data indicate PLS as the most forceful model for that discrimination (see Appendix A). Four consecutive components calculated for PCR explain 81%, while eight factors as the most important for PLS explain 95% of the variance between studied groups.

One of the most unexpected results during analyzed data was scores distribution according to gender. The PLS discrimination for cancerous and non-cancerous samples versus gender are presented in Appendix A. The scores that responded to each valuated group of SERS data, i.e., non-cancerous and cancerous, are divided by two axes: F-1 and F-2. According to the obtained results, the scores responding to the non-cancer samples collected from both women and men are located in the third quarter of the presented plots (Appendix A). A comparison of the percentages of discrimination between the samples in the distribution by cancerous and non-cancerous samples for both male and female samples and separately for men and women is shown in Figure 4. PLS prove that the first three factors explain 97%, 91%, and 61% of the variance between non-cancerous groups for men, women, and the entire population. For statistical significance, the closest values to 100% are better, and the same is true with lower numbers of concerned factors. The presented values of the first three PLS factors of the gender analysis give significantly better values than for the whole population (both genders together). These results are in line with other studies; Sikirzhytskaya et al. demonstrated that Raman spectroscopy combined with other statistic methods, such as support vector machine (SVM), might be useful in determining the sex of a blood sample [69]. It is known that biological differences between the genders result in differences in the epidemiology of cancer, i.e., in gene polymorphism and genetics/molecular level [70]. Similarly, Conforti et al. [71] noticed that when personalized immunotherapy strategies are used in advanced non-small cell lung cancer, gender is strongly associated with outcomes influencing prognosis; men and women should be treated accordingly with different and personalized immunotherapy strategies. However, so far, there are not enough experimental data focusing on this problem. In this sense, the results presented here are essential for a better understanding of the relationship between images of body fluids’ molecular structure, which may differ depending on gender. Thus, when studying neoplastic diseases from human fluids, it seems essential to consider changes within one gender.

Moreover, the analysis of the calculated PCs and Fs (for PCR and PLS, respectively) of cancerous and non-cancerous samples and type of cancer data revealed a few essential variables at 638 cm^−1^, 1142 cm^−1^, and 1452 cm^−1^ (PC1, which accounts for 48% of the total variance) and at 639 cm^−1^, 1140 cm^−1^, and 1452 cm^−1^ (F1 28% of the total variance; Appendix A). Those bands correspond to the bands observed in SERS spectra at 633 cm^−1^, 1133 cm^−1^, and 1445 cm^−1^ (Figure 2, Table 1). For tumor type differentiation in PLS analysis, F1 revealed an additional variable at 1657 cm^−1^ (1655 cm^−1^ in SERS). Those four bands are responsible for the presented differentiation between the samples of the analyzed groups.

The PLS analyses also were performed for the cancerous and non-cancerous SERS spectra in respect to the gender of the patients. The variables at 632 cm^−1^, 1127 cm^−1^, 1310 cm^−1^, and 1432 cm^−1^ for the F1 loading plots for females and only two variables at 639 cm^−1^ and 1455 cm^−1^ for men (Appendix A) were shown. Those bands are attributed to spectral regions corresponding in SERS spectra to phenylalanine (633 cm^−1^), proteins or lipids (1133 cm^−1^), collagen (1319 cm^−1^), and proteins or/and lipids (1445 cm^−1^) (see Appendix A).

Additionally, based on the predicted and correlated matrix of PCR and PLS, the receiver operating characteristic (ROC) curve was constructed for studied samples (Figure 5). To determine the correlation accuracy, specificity, and sensitivity to every possible level of probability of significance, the area under the curve (AUC) was evaluated. Those calculations were performed for whole samples of the non-cancerous and cancerous samples, squamous cell, and adenocarcinoma but also for the non-cancerous vs. cancerous samples in respect to gender. Based on the presented data calculated for non-cancerous vs. cancerous samples, the correlation accuracy for PCR is 0.67 (67%), while for the PLS-DA method, it is 0.80 (80%). The optimal cut-off point identified from the ROC curve had a sensitivity of 0.89 (89%) or 0.90 (90%) and specificity of 0.44 (44%) or 0.70 (70 %) (for PCR and PLS-DA, respectively). Presented for squamous cell vs. adenocarcinoma data (type of cancer), the correlation accuracy for PCR is 0.44 (44%), while for the PLS-DA method, it is 0.99 (99%). The optimal cut-off point identified from the ROC curve had a sensitivity of 1.00 (100%) or 0.90 (90%) and specificity of 0.60 (60%) or 1.00 (100%) (for PCR and PLS-DA, respectively). As it can be seen, in a case of PCR calculated in order to determine the type of cancer, the AUC is below 0.5. Thus, it can be surely concluded that PCR cannot be used for squamous cell vs. adenocarcinoma differentiation, as such low AUC values do not carry a medical diagnosis. In a case of samples determined from the non-cancerous vs. cancerous and in respect to gender, the correlation accuracy for PLS-DA is 0.68 (68%) for men and 0.86 (86%) for women. The optimal cut-off point identified from the ROC curve had a sensitivity of 0.67 (67%) or 0.71 (71 %) and specificity of 0.67 (67%) or 0.86 (86%) (for men and women, respectively). Presented values of sensitivity, specificity, and AUC, gathered in Table 2, definitely indicate the PLS-DA model as much better in identification of samples between the non-cancerous and cancerous, squamous cell, and adenocarcinoma pleural fluids.

To summarize, the presented method of pleural effusion analysis allows for quick and sensitive differentiation among the possible reason for pleural effusion. Furthermore, it shows a great possibility of faster diagnosis results, and finally, it gives a faster implementation of treatment, which in the case of adenocarcinoma, may significantly increase the chances of curing patients and extending their lives.

## 4. Conclusions

The potential of SERS in combination with PLS as a rapid method of diagnosing lung cancer from pleural fluids and identifying the two most common types of lung cancer was demonstrated for the first time. The PLS method is the best choice in the tested SERS data configurations (cancer vs. non-neoplastic and squamous samples vs. adenocarcinoma). However, the most visible evidence of this, apart from the percentages of the explained variance, is the calculated value of the root mean square error (RMSE) and the coefficients of determination (R2) of the calibration and prediction (e.g., RMSEC = 0.09, R2C = 0.9570 and RMSEC = 0.0553, R2C = 0.9539). In addition, the calculated area under the ROC curve is consistent with the RMSE results and confirms the effectiveness of the selected PLS method. The obtained results prove the feasibility and convenience of using the SERS method together with the PLS analysis as a competitive tool for the detection of lung cancer and even the type of cancer from the pleural fluids. Moreover, a correlation was found between the discrimination of non-cancer and cancer studies and the gender of the patients. Analysis of data in respect to gender allowed for the practical improvement of the obtained results (97%, 91% of the variance for men and women versus 68% for the entire population). This suggests that in future research, biological samples should also be analyzed, taking into account the gender of the patients of the analyzed samples.

## Figures and Tables

**Figure 1 biomedicines-10-00993-f001:**
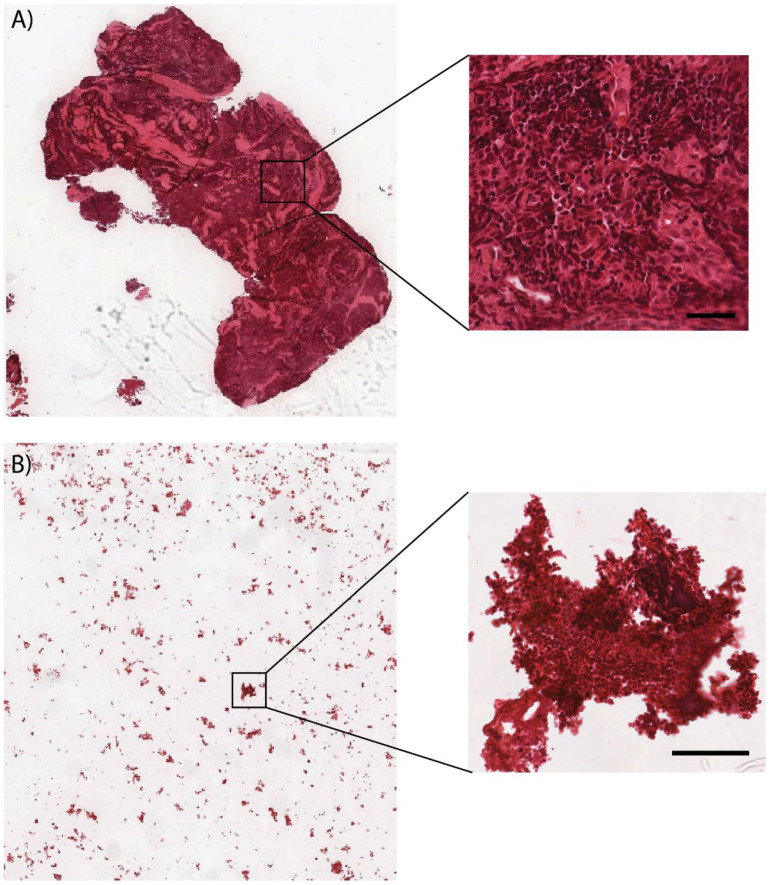
Microscopic image of pleural fluid of adenocarcinoma (**A**) and the tissue of squamous cell carcinoma (**B**) (×100, ×400, scale bar 50 µm).

**Figure 2 biomedicines-10-00993-f002:**
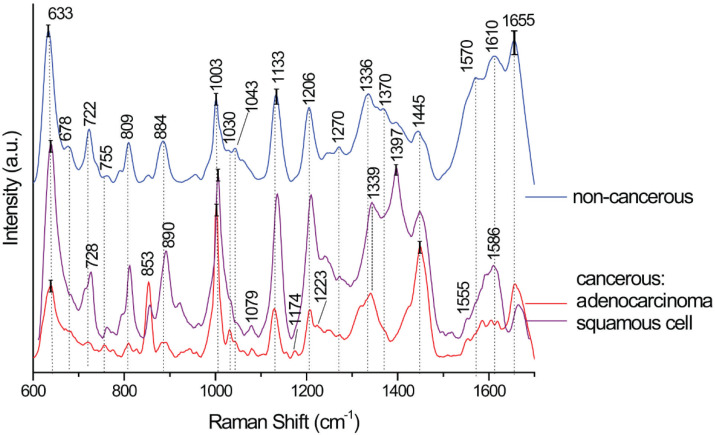
The averaged SERS spectra for non-cancerous and cancer samples for adenocarcinoma and squamous cell cancer. The bars present the SD of chosen bands.

**Figure 3 biomedicines-10-00993-f003:**
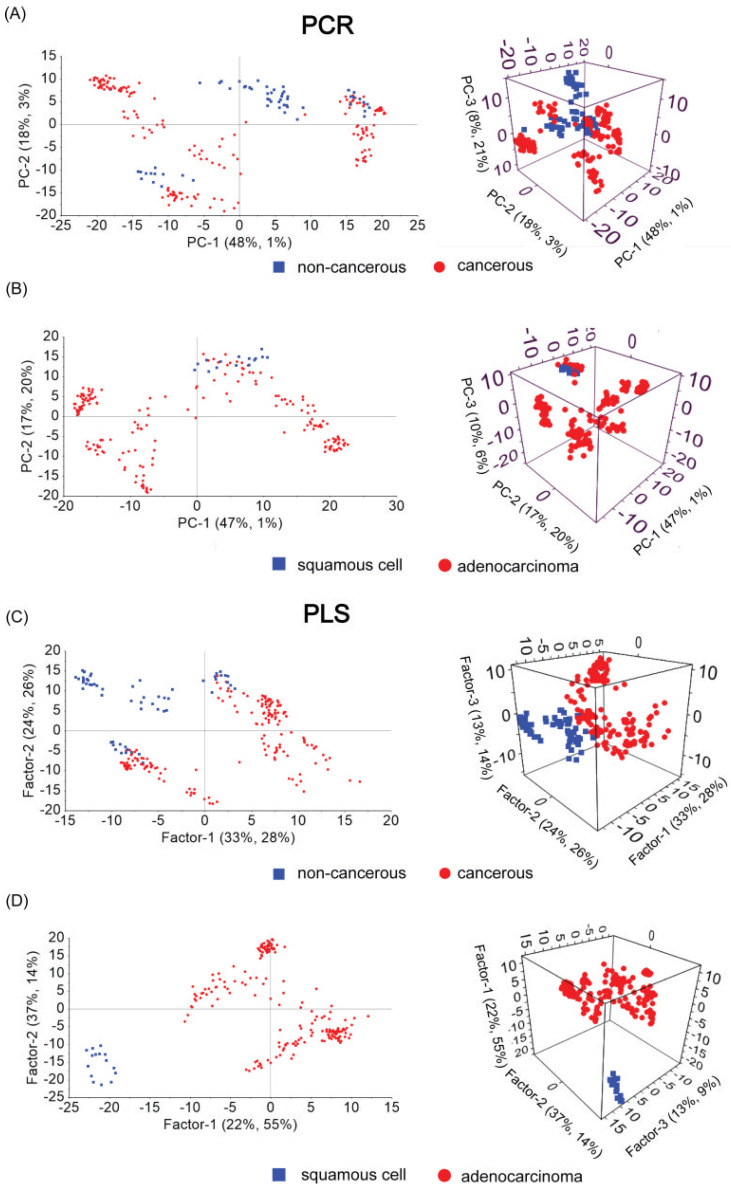
The scores plots in 2D and 3D projections for non-cancerous vs. cancerous and for squamous cell vs. adenocarcinoma samples for PCR method (**A**,**B**) and for the PLS (**C**,**D**), respectively.

**Figure 4 biomedicines-10-00993-f004:**
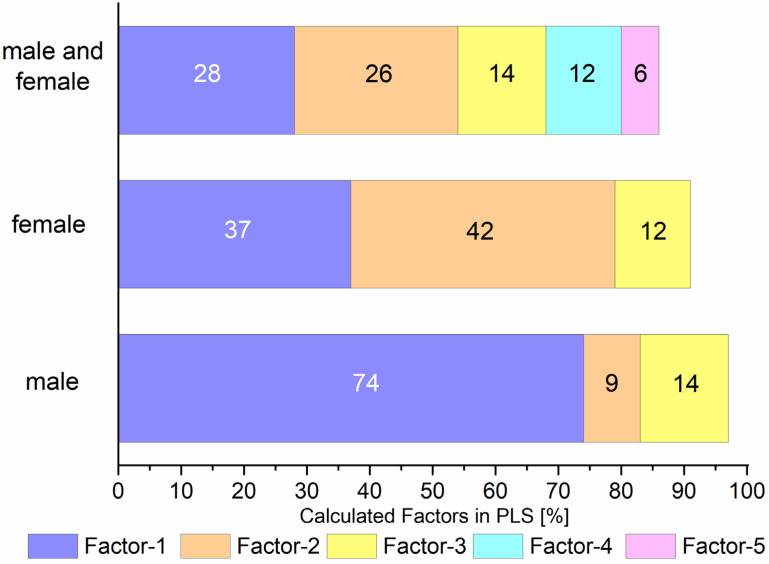
Comparisons of the first three (for male or female samples) and five PLS factors (for the entire population) in the distribution of data calculated for cancerous and non-cancerous samples. The total variance of each factor is presented on the bar in numbers.

**Figure 5 biomedicines-10-00993-f005:**
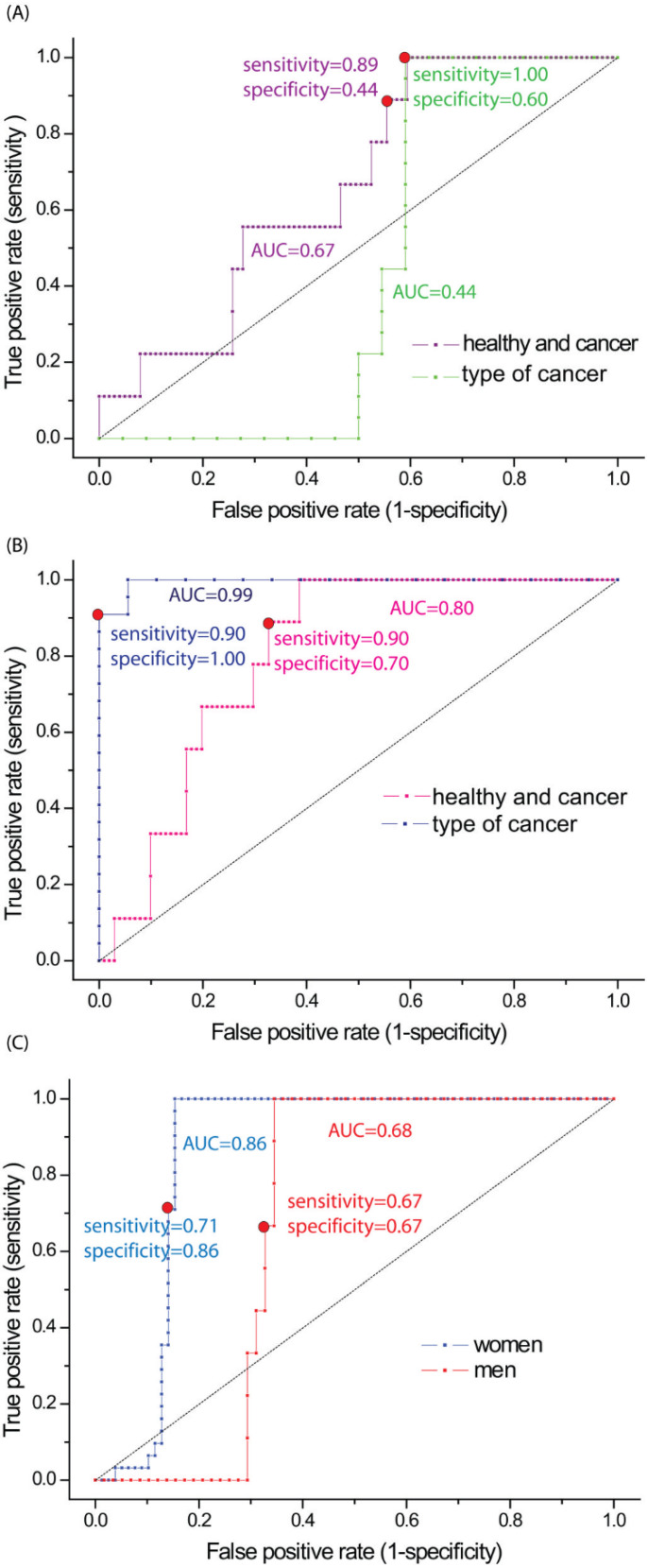
The ROC curve with the optimal operating point (red circle) for PCR method, calculated for non-cancerous vs. cancerous samples (green line) and squamous cell vs. adenocarcinoma samples (purple line) (**A**). For the PLS-DA method, calculated for the non-cancerous vs. cancerous samples (dark blue line) and squamous cell vs. adenocarcinoma samples (pink line) (**B**) and for the non-cancerous and cancerous samples for men (blue line) and women (red line) (**C**).

**Table 1 biomedicines-10-00993-t001:** Tentative assignments of bands observed in SERS spectra of adenocarcinoma and squamous cell [12,59,61,62,63,64,65,66,67,68]. The bands that show significant differences in intensity are bolded.

SERS Bands			Compound/Assignments
Cancerous		Non-Cancerous	
Adenocarcinoma	Squamous Cell		
633	633	633	Phenylalanine (skeletal)
		678	Guanine (DNA)
		722	DNA
**728**	728		Tryptophan, lipids
755		755	CH2 rocking, symmetric breathing, tryptophan
809	809	809	Cytosine, uracil, tyrosine
**853**	**853**		Tyrosine, proteins
**890**	890	884	Proteins
1003	1003	1003	Phenylalanine (ring breathing mode)
1030	1030	1030	Proteins, C-H in plane Phe, deoxyribose, str. (C-O)
1043		1043	Proteins, ν (C-O), ν (C-N)
**1079**	**1079**		CC or PO_2_ stretching, phospholipids in nucleic acids
1133	1133	1133	ν (C-N) of proteins or ν (C-C) lipids
**1174**			CC stretching, L-phenylalanine, proteins
1206	1208	1206	N-C-C stretching and bending
1223			Amide III
1270	1270	1270	
1319			CH_3_ def. in collagen
1339	1339	**1336**	Adenine ring breathing, phospholipids, or nucleic acid
		**1370**	Guanine in DNA/TRP (protein)/lipids
	1397		CO of the COH stretching of amino acids in proteins or COO stretching
1445	1445	1445	CH_2_ bending in proteins and lipids, keratin, fatty acids, triglycerides, CH_2_, CH_3_ deformation/lipids/proteins C–H wag.
**1555**	**1555**		Guanine in DNA/adenine/TRP (protein)
		1570	
	1586		
	1610	1610	Phenylalanine, tyrosine, cytosine,
**1655**	1660	1655	Amide I/C=C lipid stretch

**Table 2 biomedicines-10-00993-t002:** Results for PCR and PLS-DA classification model for non-cancerous vs. cancerous samples and squamous cell vs. adenocarcinoma samples.

Type of Tested Samples	Method	Sensitivity	Specificity	AUC
non-cancerous vs. cancerous	PCR	0.89	0.44	0.67
	PLS-DA	0.90	0.70	0.80
Women	PLS-DA	0.71	0.86	0.86
Men	PLS-DA	0.67	0.67	0.68
squamous cell vs. adenocarcinoma	PCR	1.00	0.60	0.44
	PLS-DA	0.90	1.00	0.99

## Data Availability

Not applicable.

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
