# Peer review of "Lung Cancer: Spectral and Numerical Differentiation among Benign and Malignant Pleural Effusions Based on the Surface-Enhanced Raman Spectroscopy"

_biomedicines, 2022, doi:10.3390/biomedicines10050993_

Round 1
Reviewer 1 Report
The paper entitled "Lung cancer: spectral and numerical differentiation among benign and malignant pleural effusions based on the surface-enhanced Raman spectroscopy" proposes a model to discriminate malignant and normal lung conditions using SERS with the additional possibility of discriminating the type of cancer. The work is sound and interesting, however in my opinion some issues should be addressed:
- The work requires careful proofreading and a general improvement in the language, especially in the discussion where English and sentence construction problems sometimes hinder comprehension.
- Lines 256-260: It is not clear what criteria the authors decided to include all spectra in the analysis. Precisely because small differences in spectra can be significant, it is important to ensure high data quality. Did the author consider applying a minimum intensity filter? Please rephrase the sentences to better clarify.
- Figure 4: it is not clear to me what message this figure is trying to convey. Could the author better clarify the diagnostic significance of a different number of factors? Please also rephrase the caption as the meaning of the numbers in the colored bars is expressed in a confusing way.
- Lines 304-308 and 365-369: Data reported indicated that PLS factors related to gender analysis give better values than for the whole population as the first three factors account for 97% and 91% of the variance for males and women respectively, versus only 68% for male + women. However, gender-based sensibility and sensitivity of the method resulted lower with respect to the whole population. Could you please comment on this?
- I suggest improving the quality of the images, as to meet the journal standards.
Author Response
Reviewers' comments:
Reviewer 1
The paper entitled "Lung cancer: spectral and numerical differentiation among benign and malignant pleural effusions based on the surface-enhanced Raman spectroscopy" proposes a model to discriminate malignant and normal lung conditions using SERS with the additional possibility of discriminating the type of cancer. The work is sound and interesting, however in my opinion some issues should be addressed:
1. The work requires careful proofreading and a general improvement in the language, especially in the discussion where English and sentence construction problems sometimes hinder comprehension.
Thank you very much for your comment. We have corrected all typos along the manuscript.
- Lines 256-260: It is not clear what criteria the authors decided to include all spectra in the analysis. Precisely because small differences in spectra can be significant, it is important to ensure high data quality. Did the author consider applying a minimum intensity filter? Please rephrase the sentences to better clarify.
Thank you very much for your comment.
Indeed, to ensure the high quality of the obtained data and according the information given in the Experimental section the raw spectra were processed by vector normalization, smoothing, and baseline correction to avoid errors or artificial interference during the sample preparation and spectral acquisition.
We have changed this sentences to make more clear:
“The rest of the PCs or Fs important for calculated discrimination are presented in Figure 2S and 3S (Supplementary Information), respectively. It should be mentioned that average of the standard deviation (SD) variation up to 15% (Figure S1; Supplementary Information) are obtained for the recorded spectra. Therefore, all gathered spectra, recorded for one sample in the mapping mode, were analyzed and included in the presented classification.”
- Figure 4: it is not clear to me what message this figure is trying to convey. Could the author better clarify the diagnostic significance of a different number of factors? Please also rephrase the caption as the meaning of the numbers in the colored bars is expressed in a confusing way.
Thank you very much for your comment.
We have changed the caption of Figure 4, as well as, the axis X labelling and rephrased the information concerning this figure:
“PLS prove that the first three factors explain 97%, 91%, and 61% of the variance between non-cancerous groups for men, women, and the entire population. For statistical significance the most close values to 100% is better, as well as lower number of concerned factors. The presented values of the first three PLS factors of the gender analysis, give significantly better values than for the whole population (both genders together).”
[here improved Figure 4]
“Figure 4. Comparisons of the first three (for male or female samples) and five PLS factors (for the entire population) in the distribution of data calculated for cancerous and non-cancerous samples. The total variance of each factors are presented onto bar as number.“
- Lines 304-308 and 365-369: Data reported indicated that PLS factors related to gender analysis give better values than for the whole population as the first three factors account for 97% and 91% of the variance for males and women respectively, versus only 68% for male + women. However, gender-based sensibility and sensitivity of the method resulted lower with respect to the whole population. Could you please comment on this?
Thank you very much for your comment.
As it is presented in the manuscript:
„first three PLS factors of the gender analysis, give significantly better values than for the whole population (both genders together).”
As it gives 97% and 91% for men and women considered separately, versus 61% for the entire population. Those values was also very interesting for us, and especially because are consistent with other already published work. Thus we decided to calculate AUC, specificity and sensitivity also for separated data in respect with gender. And as it is presented in Table 2, the correlation accuracy for PLS-DA calculated for men is 0.68 (68%), the optimal cut-point identified from the ROC curve had a sensitivity of 0.67 (67%) and specificity of 0.67 (67%). Thus it is lower than those values obtained for women: the correlation accuracy 0.86 (86%), sensitivity 0.71 (71 %), and specificity 0.86 (86%). Interestingly, higher values of PLS of the first three factors lower AUC, specificity and sensitivity. Moreover, those values calculated for the entire population: AUC 0.67 (67%), sensitivity 0.89 (89 %), and specificity 0.44 (44%). In other words, for the gender analysis the obtained values prove the lower statistical significance.
However, it should be in mind, that the calculation made for the entire population were based on SERS data recorded for 6 men and 6 women samples (12 in total; 180 spectra). And only half of those samples, were used, for gender analysis calculation (PLS-DA based on 90 spectra for either men or women). Most probably, with increasing the number of samples more spectra will be considered and better statistical significance should be obtained. Thus, as it is written in the discussion part:
"In this sense, the results presented here are essential for a better understanding of the relationship between body fluids image of molecular structure, which may differ in dependence on gender. Thus, when studying neoplastic diseases from human fluids, it seems essential, to consider changes within one gender."
- I suggest improving the quality of the images, as to meet the journal standards.
Thank you very much for your comment.
We have found that Figure 3, by mistake, was lower quality (150 DPI). We have improved this problem in the revised version of the manuscript.
Reviewer 2 Report
This is a nicely planned study on the potential use of surface-enhanced Raman spectroscopy to spectrally and numerically differentiate benign and malignant pleural effusions from lung cancer patients. The study is straightforward, and the results support the interpretations and the conclusions. Sections are well written. Similar studies, however, have been performed for various cancers, but not on pleural effusions of lung cancer patients. In lung cancer patients, a similar approach has been performed in the saliva samples of patients.
Since performing this technique in saliva samples is a much more practical, cost- and time-effective, and non-invasive approach, this paper does not discuss the saliva-based studies and justify why analysis of pleural effusions is superior (or inferior) to the known approach to saliva samples. If indeed this study is superior, a justification for this approach and its advantages over the other methods must be discussed in the manuscript.
How does the data compare between this study and the previous? Are the parameters the same or different?
Overall, the study is well done. Some discussions positioning this study in line with the literature and describing the limitations of the study, if any, would strengthen the manuscript.
Author Response
Reviewer 2
This is a nicely planned study on the potential use of surface-enhanced Raman spectroscopy to spectrally and numerically differentiate benign and malignant pleural effusions from lung cancer patients. The study is straightforward, and the results support the interpretations and the conclusions. Sections are well written. Similar studies, however, have been performed for various cancers, but not on pleural effusions of lung cancer patients. In lung cancer patients, a similar approach has been performed in the saliva samples of patients.
Since performing this technique in saliva samples is a much more practical, cost- and time-effective, and non-invasive approach, this paper does not discuss the saliva-based studies and justify why analysis of pleural effusions is superior (or inferior) to the known approach to saliva samples. If indeed this study is superior, a justification for this approach and its advantages over the other methods must be discussed in the manuscript.
How does the data compare between this study and the previous? Are the parameters the same or different?
Thank you very much for your comment.
Indeed, X. Li et al (DOI: 10.1117/1.JBO.17.3.037003) present results for SERS saliva study, in which, they differentiating between cancer and healthy control patients. To find the differences between healthy and carcinoma patients they calculated the difference spectrum. Most of the Raman peak intensities decrease for lung cancer patients compared with that of healthy one. The peaks were assigned to proteins and nucleic acids, which indicate a corresponding decrease of those substances in saliva. To reduce and discriminate between the two groups of data the unsupervised multivariate methods have been used, such as Principal component analysis (PCA) and linear discriminant analysis (LDA).
Contrary to the work presented above, in the SERS data of pleural fluids (both cancerous and noncancerous samples) the new bands in the SERS spectra of cancer patients were revealed, assigned to the vibration of proteins, amide III, and Guanine in DNA/adenine/TRP (protein).
In our study we used the chemometric supervised techniques of classification (partial least squares - PLS) and quantification (principal component regression - PCR) of SERS results. This is evidenced by the calculated values of the root mean square errors of calibration and prediction, but also the coefficients of calibration determination and prediction (R2C = 0.9570 and R2C = 0.7968), which are more robust and ruggedness compared to those calculated for PCR. In addition, the relationship between cancerous and non-cancerous samples in the dependence on the gender of the studied patients is presented.
There is no way to compare the method of analysis two different samples such as pleural fluid and saliva. The SERS spectra of saliva is nonspecific and any changes observed may be associated with many diseases, inflammations, including cancer and metastases. Currently, based on our experience in saliva tests (SERS based methods), we know, the image of saliva is influenced by the method of its collection, storage conditions and preparation for the test itself. Therefore studies of saliva are not conclusive and specific. However, the analysis of the saliva SERS spectra may help in screening tests of infected people - they will not be tests that indicate a specific disease. Contrary, SERS-based tests of pleural fluid, compete with cytology and these are tests that will clearly indicate a given disease. In a situation where pleural effusion is causing respiratory symptoms and requires drainage, diagnosis of SERS fluid will be faster than standard cytology. Additionally, in the case of methods presented in our article, the supervised numerical methods are used to analyze data to be used in the future for clinical purposes (unsupervised, as in the X. Li et al article are not).
Overall, the study is well done. Some discussions positioning this study in line with the literature and describing the limitations of the study, if any, would strengthen the manuscript.
Thank you very much for your comment.
We tried to refer in our research and analysis to the available literature and to discuss (as much as we could) the advantages and disadvantages of the techniques we used.